



# Sub-seasonal forecasts of demand, wind power and solar power generation for 28 European Countries

Hannah C. Bloomfield[1], David J. Brayshaw[1,2], Paula L. M. Gonzalez[1,2,3], and Andrew Charlton-Perez[1]

[1]University of Reading (UK)
[2]National Centre for Atmospheric Science, Reading (UK)
[3]International Research Institute for Climate and Society, The Earth Institute, Columbia University, Palisades, New York, (USA)

**Correspondence:** (h.c.bloomfield@reading.ac.uk)

**Abstract.** Electricity systems are becoming increasingly exposed to weather. The need for high-quality meteorological forecasts for managing risk across all timescales has therefore never been greater. This paper seeks to extend the uptake of meteorological data in the power systems modelling community to include probabilistic meteorological forecasts at sub-seasonal lead-times. Such forecasts are growing in skill and are receiving considerable attention in power system risk management and energy trading. Despite this interest, these forecasts are rarely evaluated in power system terms and technical barriers frequently prohibit use by non-meteorological specialists.

This paper therefore presents data produced through a new EU climate services program Subseasonal-to-seasonal forecasting for Energy (S2S4E). The data corresponds to a suite of well-documented, easy-to-use, self-consistent daily- and nationally-aggregated time-series for wind power, solar power and electricity demand across 28 European countries. The DOI http://dx.doi.org/10.17864/1947.275 will be activated after the paper has been accepted for publication. In the meantime, the data is accessible via https://researchdata.reading.ac.uk/275/, (Gonzalez et al., 2020). The data includes a set of daily ensemble reforecasts from two leading forecast systems spanning 20-years (ECMWF, 1996-2016) and 11-years (NCEP, 1999-2010). The reforecasts containing multiple plausible realisations of daily-weather and power data for up to 6 weeks in the future.

To the authors' knowledge, this is the first time fully calibrated and post-processed daily power system forecast set has been published, and this is the primary purpose of this paper. A brief review of forecast skill in each of the individual primary power system properties and the composite property demand-net-renewables is presented, focusing on the winter season. The forecast systems contain additional skill over climatological expectation for weekly-average forecasts at extended lead-times, though this skill depends on the nature of the forecast metric considered. This highlights the need for greater collaboration between the energy- and meteorological research communities to develop applications, and it is hoped that publishing these data and tools will support this.

## 1 Introduction

A key feature of current large-scale power systems is that the demand for electricity must be met by electricity generation on a near instantaneous basis. Historically, this has been achieved by scheduling a combination of coal, gas and nuclear power sta-





tions to meet a forecast demand, which is strongly dependent on temperature (Bessec and Fouquau, 2008). The growing use of

wind and solar-PV generation, however, leads to new challenges. The generation output from these weather-dependent sources

is determined by meteorological conditions, and thus cannot be controlled to the same extent as the generation from traditional

power plants. Both demand and generation-potential therefore contain strongly weather-sensitive components. Given the ne-

cessity of ensuring the balance between electricity production and demand, an accurate estimation of future weather state can

improve the efficiency and reliability of energy management at local and national scales, and provide a more realistic estimate

of future energy prices.

The impact of a shift to weather-sensitive generation has implications not only for the owners and operators of renewable

resources, but across the power system. Skillful forecasts of country-aggregated demand and renewable generation are believed

to provide valuable contextual information to a variety of energy system stakeholders: from individual traders, power plant

operators and owners, to national transmission system operators (White et al., 2017; Soret et al., 2019). Although the use

of short-range weather forecasts is now common in the energy sector and there has been a substantial amount of academic

literature on the topic (Bossavy et al., 2013; Füss et al., 2015; Drew et al., 2017; Cannon et al., 2017; Browell et al., 2018;

Stanger et al., 2019). There has been comparatively little attention paid to the use of sub-seasonal to seasonal (S2S) forecasts

by energy users for decision making. This is possibly conistent with the perceived difficulty of extracting predicsignals from

extended range forecasts (Soares and Dessai, 2016). However, recent advances in forecasting have begun to result in skillful

longer range predictions for: European demand (De Felice et al., 2015; Clark et al., 2017; Thornton et al., 2019; Dorrington

et al., 2020), wind power generation (Lynch et al., 2014; Beerli et al., 2017; Soret et al., 2019; Torralba et al., 2017; Lledó et al.,

2019; Bett et al., 2019; Lee et al., 2019), solar power generation (Bett et al., 2019) and Hydro power generation (Arnal et al.,

2018) which can consequently lead to improvements in awareness, preparedness and decision-making from a user perspective

(Goodess et al., 2019).

It has previously been highlighted that there are a number of technical barriers to the initial uptake of S2S forecasts by

users including understanding the inherent uncertainty (Soares and Dessai, 2016). To the best of the authors' knowledge fully

calibrated and post-processed daily power system data has not been published for use in energy systems research applications.

The aim of this paper is therefore to describe a new open access dataset of national demand, wind power and solar power

forecasts created to explore and demonstrate the usefulness of sub-seasonal predictions to the energy sector. Section 2 describes

the methods used to build these country-level forecasts, gives details of the S2S prediction systems used and the skill scores

used to validate the models. Section 3 gives examples of the skill present within the forecasts using a variety of time periods,

variables and skill scores, concluding with some illustrative case studies. Concluding discussion is given in section 4.

## 2   Methods

This section briefly describes the meteorological reanalysis (section 2.1) and sub-seasonal forecast systems (section 2.2) used

in this study. Following this the methods to convert both reanalysis and forecast data into estimates of national weather-

dependent demand (2.3), wind power generation (2.4) and solar power generation (2.5) are briefly described. Each model





converts meteorological data to energy variables at the highest possible spatial and temporal resolution available. As these conversion methods represent only minor modifications to methods that have been presented before at length in Bloomfield et al. (2020b), readers are referred to Appendices A-C for a full description of how hourly time series of demand, wind and
solar power are created. After these have been calculated demand can be subtracted from wind power generation to obtain demand-net-wind. Similar calculations can be done for demand-net-solar and demand-net-renewables but these are excluded here for brevity.

## 2.1 The ERA5 reanalysis

The meteorological data used for validation in this study is from the ERA5 reanalysis (Hersbach et al., 2020), available from
CDS (2020). A reanalysis is a reconstruction of the recent atmosphere, which is created through running a numerical weather prediction model, with data assimilation to ingest all available observations for a given period. The ERA5 re-analysis is currently available from 1979-present, in hourly time steps at $0.3°$ spatial resolution. The S2S forecast models are however, only available at lower spatial and temporal resolution than the ERA5 reanalysis through the S2S archive, see Vitart et al. (2017). Therefore, a reduced resolution version of the ERA5 reanalysis has been created, and is used throughout in this study, which
has only daily values and $1.5°$ spatial resolution. National 2m temperature and surface solar radiation, as well as gridded 100m wind speed data at this resolution are available to download alongside the energy variables discussed in this study for those wishing to explore the predictability of meteorological variables behind the energy models.

A point to note is that for accurate representation of the National wind power generation a mean bias correction procedure was required to correct ERA5 to the Global Wind Atlas dataset (GWA, 2018), as in Lledó et al. (2019). This is to correct the
anomalously low 100m wind speeds found over European land compared to those seen in the Global Wind Atlas.

## 2.2 Sub-seasonal reforecasts

Sub-seasonal forecasts from the ECMWF and NCEP models have been obtained from the S2S database for analysis in this study (available at: https://apps.ecmwf.int/datasets/data/s2s/, see Vitart et al. (2017) for a description of the database). In the S2S database both real-time forecasts (i.e., the version of the model that the forecast provider ran to give predictions) and
reforecasts are available. The reforecasts are a set of forecasts which accompany the real-time forecasts, using the same model and initialised on the same day of the year for the preceding 10 to 20 years. The reforecast data is typically produced to allow forecasts to be corrected or re-calibrated for the effects of "model drift" (with the drift being evaluated over the re-forecasts for which there are corresponding observational recordings). By comparing the set of reforecasts to a reanalysis, or other available observations (as in Monhart et al. (2018)), the model drift can be determined and the forecast can be corrected. The reforecasts
are also a useful tool to assess the forecast model performance, as they cover many more years and more meteorological conditions than the real-time forecasts. Here, we focus exclusively on the re-forecast components, using a "leave-one-year-out" strategy for re-calibration (see below).

The different sub-seasonal prediction centers do not have a common forecasting strategy. The ECMWF model produces two forecasts a week (Mondays and Thursdays) where an 11 member ensemble is available in the reforecasts. The reforecast period





used here is the 20 years from 1996-2015 from cycle CY41R1. The NCEP model produces a forecast every day, but only with four ensemble members. To make the fairest possible comparison between these two models a lagged-ensemble is constructed from NCEP model version T126L64GFS. To do this the matching reforecasts from Mondays and Thursdays are taken from the NCEP reforecasts (as available in the ECMWF model) and are combined with reforecast launched on the 2 preceding days to provide a 12 member ensemble twice a week: a so-called lagged ensemble. The first set includes Saturday, Sunday and

Monday reforecasts and the second including Tuesday, Wednesday and Thursday forecasts.

The fields available from the S2S database are daily-mean 2m temperature and midnight 10m wind speed at 1.5° spatial resolution. Before the energy variables are calculated, the sub-seasonal forecasts of 2m temperature, 100m wind speed and surface solar irradiance are bias corrected using a corresponding 1.5 degree version of ERA5 as the reference. We note that as only 10m wind speeds are available from the S2S database these are extrapolated first to 100m using a power law:

$$U_{100m} = U_{10m}(\frac{Z_{100}}{Z_{10}})^{\alpha} \qquad (1)$$

Here U is the wind speed, Z is the height from the surface and $\alpha = \frac{1}{7}$. This method is commonly used in the wind power modelling community.

The surface solar radiation output from the S2S database is a daily output aggregated since the start of the forecast (i.e., the 5th daily value recorded in the forecast is the accumulated radiation over a 5-day period) and had to be differentiated temporally

to obtain daily accumulations. This temporal differentiation however, produces isolated occurrences of small negative values in the resulting daily surface solar irradiance. As negative values are clearly nonphysical these are transformed into null values.

Each individual reforecast can be corrected by comparing all the other contemporaneous reforecasts to a reanalysis. This sometimes known as a leave-one-out approach (e.g. 1996 is corrected using data from 1997-2015). The method of bias correction used in this study is variance inflation. This is described in detail in Doblas-Reyes et al. (2005) and used in Torralba et al.

(2017) and Lee et al. (2019). The method ensures that the reforecast mean and variance agrees with those in ERA5, and also that the correlation between the reforecast and ERA5 is preserved (Doblas-Reyes et al., 2005).

A subtlety occurs, however, in winter months for the northernmost countries (Sweden, Norway and Finland) which are affected by the Polar night and receive little to no solar radiation. A standard application of the bias inflation methodology (Doblas-Reyes et al., 2005) result in non-physical values in this situation (as the forecast has zero (or very little) spread during

the polar night). The instances in which this occurred, in winter months within the Arctic circle, were identified and the bias correction was reverted to a standard lead-dependent mean correction over those points.

The bias corrected, gridded, meteorological variables are converted to national energy demand, wind and solar power generation using the models described in Sections 2.3, 2.4 and 2.5, respectively. National average bias corrected 2m temperature, and surface solar radiation data, as well as gridded 100m wind speeds from the reforecasts are nevertheless made available to down-

load alongside the energy variables discussed in this study for those wishing to explore the predictability of meteorological variables behind the energy models.





The reforecasts for the aforementioned variables are provided alongside the corresponding ERA5 parameters, in a structure that matches the forecast design. This facilitates any further comparison or verification of the products. Additionally, the full set of ERA5-based variables, derived using the hourly energy conversion models at the reanalysis native resolution and for the period 1979-2019 is also provided (Bloomfield et al., 2020a).

## 2.3 Demand Model

The demand model used in this study is intended to capture the weather-dependent fluctuations in national demand. It is constructed in two stages. Firstly a multiple linear regression model is constructed for observed national demand (2016-2017, from the ENTSOe transparency portal, ENTSOE (2018)) The regression technique includes both weather-dependent and human-behaviour dependent factors. The weather-dependent model parameters are Heating-Degree days (HDD), and Cooling-Degree Days (CDD) which are a non linear function of country-average temperature (see Appendix A for a mathematical definition). HDD are relevant for countries in which electricity is required for heating (generally in winter) whereas CDD are relevant for Countries which require electricity for cooling (predominantly in summer). This full model (including both human behavioural and weather factors) performs well when validated on real data (average daily $R^2$ of 0.80 with average percentage error of 7%, Bloomfield et al. (2020b)).

For this application we are only interested in the weather-dependent component of demand, for which the sub-seasonal forecasts provide predictions. In the second step, we therefore choose to remove all the human behavioural factors (such as long-term trends and day-of-week effects) by setting the corresponding regression coefficients to zero (see Appendix A for further details of the model). The resulting model is therefore interpreted as an estimate of the expected national demand given a set of input meteorological conditions (i.e., HDD and CDD) in the absence of confounding human factors.

## 2.4 Wind Power Model

A physical model is used to produce estimates of national wind power generation. Gridded 100m wind speeds are converted into wind power capacity factors using an appropriate wind-turbine power curve, which is selected using the bias adjusted 1.5° resolution version of ERA5 1979-2019 mean 100m wind speed in each grid box. This optimises the most appropriate class of turbine (see Appendix B for details) to be built in each grid box to maximise the potential wind power generation. The resulting capacity factors are multiplied by the estimated installed capacity in each grid box and aggregated over the country domain. Information regarding the spatial distribution and installed capacity of wind turbines is taken from $thewindpower.net$ database. Validation of this model on the native ERA5 grid is available in Bloomfield et al. (2020b). The models perform well compared to others in the literature, with an average daily $R^2$ of 0.91, and average percentage error of 10% when validated against data from ENTSOE (2018), see Appendix B for further details.





## 2.5 Solar Photovoltaic (PV) Model

The solar PV model follows the empirical formulation of Evans and Florschuetz (1977) but with adaptation to newer solar PV technologies using methods from Bett and Thornton (2016). The meteorological inputs are grid point temperature and incoming surface solar irradiance (G), from which national solar power capacity factor is calculated using the equation below:

$$CF(t) = \frac{power}{power_{STC}} = \eta(G,T)\frac{G(t)}{G_{STC}(t)} \tag{2}$$

Where G is the incoming surface solar radiation and T is the grid box 2m temperature, and t is the time step (days). STC stands for standard test conditions (T = 25°C G = 1000 Wm$^{-2}$) and $\eta$ is the relative efficiency of the panel following"

$$\eta(G,T) = \eta_r[1 - \beta_r(T_c - T_r] \tag{3}$$

Where $\eta_r$ is the photovoltaic cell efficiency evaluated at the reference temperature $T_r$, $\beta_r$ is the fractional decrease of cell efficiency per unit temperature increase and $T_c$ is the cell temperature (assumed to be identical to the grid box temperature). The model performs well with an average daily R$^2$ of 0.93 with 3.2% error for countries where data was available at high enough quality for validation from the ENTSOe transparency portal ENTSOE (2018). Further details of the model formulation are given in Appendix C.

## 2.6 Verification Metrics

A forecast of daily energy demand or wind power at a lead time of two or three weeks is unlikely to provide an exact representation of daily conditions. One may hope, however, to see some skill in forecasting weekly means. This paper therefore assesses the skill of the means of the energy variables over four consecutive weeks of forecast lead time starting on day 5 (where day 1 consists of the first 24 hours of the forecast). Under this convention, weeks $1 - 4$ of the forecast encompass days $5 - 11$, $12 - 18$, $19 - 25$, and $26 - 32$ respectively. The reason for starting at day 5 is that at lead times less than 5 days is not the focus of this study, or the intended use of the S2S forecasts.

In this study the energy variables calculated from the ERA5 reanalysis are considered as truth. This allows the potential value of the sub-seasonal models in predicting weather-dependent fluctuations in demand, wind and solar power generation to be clearly assessed. This study does not try to account for human-induced changes such as maintenance schedules of plants, system driven curtailment or public holidays. All forecasts used in this study assume a set weekday parameter (Monday).

Three verification metrics are used to assess the performance of the ensemble forecasts of energy variables. The first metric assessed is the anomaly correlation coefficient (ACC) of the ensemble mean of the reforecast. The other two metrics treat the ensemble forecasts as probability forecasts (Wilks, 2011). The first probabilistic metric was the Ranked Probability Skill Score (RPSS) of the three events determined by the terciles of the distribution of the variable. This assesses the performance of the forecast when the continuous variable is reduced to three categories (below normal, normal, above normal, Epstein (1969)). The



second probability metric was the Continuous Ranked Probability Skill Score (CRPSS). This assesses the forecast probability distribution of the continuous variable (Brown, 1974). The two probabilistic skill scores give the skill of the sub-seasonal forecast relative to a climatological forecast that always forecasts the climatological probabilities of the events or categories involved.

An important point to note is that the dataset described here presents only reforecast data. As reforecasts typically have fewer ensemble members than operational forecasts, this tends to limit the skill of reforecast relative to a true forward-looking forecast. In the case of the S2S ECMWF forecasts, the real-time forecasts have 51 ensemble members whilst the reforecasts have only 11. To account for this the method of Ferro et al. (2008) is applied. In calculating the two probability metrics, such a correction was applied for the forecast, but not for the climatological benchmark forecast.

## 3 Results

Within the dataset which we have published there are 6048 ($= 2 * 28 * 12 * 3 * 3$) combinations of forecast model, country, month, forecast energy variable, and verification metric which could be discussed. The results in this section therefore present only a general overview of the skill of the dataset. We encourage others to fully investigate time periods, events and skill scores most relevant to their application. The results presented concentrate on six representative countries: France, Germany, Sweden, Romania, Spain, and the United Kingdom, which are chosen due to their geographic diversity and varying power system composition.

### 3.1 Variations in skill in forecasting demand and wind power generation throughout the year

Figure 1 shows the typical seasonal cycle of skill in a range of forecast properties. It is immediately clear that the sub-seasonal forecasts generally contain a good level of skill for week 1 (forecast days 5-11) throughout the year. For all of the energy variables in question greater skill is seen for the ECMWF model than the NCEP model. It is, however, important to contextualise this result by noting that the *truth* used to validate the forecasts is based on an ECMWF product (ERA5) and shares a model heritage with the ECMWF forecast system. This is likely to be associated with a weaker relative performance in the NCEP system, in addition to factors such as the formation of the lagged-ensemble and potentially lower skill inherent in the NCEP model itself.

The largest amount of skill is seen for demand in the ECMWF model, with ACCs of around 0.8 for all of the case study countries during the winter period (see Figure 1a). Skill is present throughout the year, though it is slightly weaker in summer than winter, particularly for Germany and Spain. Similar results are seen for the skill in forecasting ECMWF wind power generation (Figure 1c), although the maximum level of skill is lower. This suggests either lower skill in forecasting wind speeds than temperatures, or may be associated with subtleties in the method of creating the energy variables. (E.g., the demand model averaging temperatures over the whole country prior to conversion to demand, therefore allowing for some aggregation of skill. Whereas wind power is estimated on a grid point basis prior to country-aggregation.) Generally the skill in demand-net-wind for the ECMWF model is lower than the skill of demand, but higher than that of wind (Figure 1b). The reduction in skill





is dependent on the amount of wind power generation installed in the country. The ECMWF model has a similar (if slightly reduced) skill in forecasting the solar power generation compared to wind power generation in the countries shown in Figure 1d. The skill is generally uniform throughout the year, with some exceptions in the summer months.

The results for the NCEP model (Figure 1e-h) are similar to those for the ECMWF model but with a generally lower level of skill. The reduction in skill during summer is also much larger for demand and demand-net-wind in the NCEP model compared to the ECMWF model.

In both seasons there is a relatively uniform skill distribution over Europe. However, there are exceptions, with Romania seeing generally low levels of skill compared to the other countries. This could suggest difficulties in converting weather

variables to power variables in some locations. As skill is generally highest in winter we choose to focus on this period for the analysis in following sections.

### 3.2    Deterministic Skill Assessment

Figure 2 shows the ACC of the three energy variables for the selected case study countries plotted against forecast lead time, for weeks 1-4 in January. For forecasts of demand from the ECMWF model (Figure 2a) a steep decline in skill is seen between

week 1 and week 2 (days 12-19), with a maximum ACC of 0.5 seen for Romania. Positive values of ACC are still seen out to week 4 for Romania, however most countries only show significant positive ACC in week 3. Similar results are seen for demand-net-wind (Figure 2b) wind power generation (Figure 2c ) in the ECMWF model, while for the NCEP model (Figure 2e-h) there are slightly reduced levels of skill (as seen in Figure 1). The differences in skill between the ECMWF and NCEP models are less as obvious in weeks 2, with larger difference in weeks 3-4. Qualitatively similar results are seen for all of the

extended winter months from November to March (not shown).

An interesting observation is the high level of skill seen in January Solar power forecasts for Sweden in both models compared to the other countries chosen (Figure 2d,h). However, we note that this relatively high forecast skill is likely due to the polar night which impacts upon Scandinavian countries (i.e., the forecast is simply predicting that the occurrence of no solar radiation due to the polar night). This high skill phenomena is similar to the high skill in precipitation forecasts seen in

dry regions (e.g. in Li and Robertson (2015) a dry mask is used to combat the impact of high precipitation predictability in dry regions). This highlights the need for careful assessment of forecast skill when used in energy applications.

This shows the potential for sub-seasonal forecasts to provide useful information on weekly-mean power system operation out to at least 2 weeks ahead (19 days in this case), with information at 4 weeks lead time in some countries. The anomaly correlation coefficient is, however, a very simple deterministic measure of skill, providing only an indication of whether the

weekly mean of a variable is correctly predicted to be high or low relative to a baseline average. More detailed probabilistic information may be desirable for decision making. The potential for this is examined in the next section.

### 3.3    Probabilistic forecast skill indicators

Figure 3 shows forecast skill for winter (December-February) demand-net-wind present in week 1-4 for the ECMWF model, when comparing three different metrics of skill across Europe: the deterministic score ACC (as discussed in the previous





section) along with probabilistic scores RPSS and CPRSS. Similar results are seen for the NCEP model (not shown). In general, the level of skill degrades as the complexity of the information required from the forecast increases. For example there is much more skill in week 1 when forecasting the tercile of electricity demand (Figure 3e) than for trying to forecast a well-resolved cumulative frequency distribution of demand from the ensemble (Figure 3i). Spatial differences in skill also become larger with increasing complexity of the metric, suggesting a high sensitivity of the forecast skill to the specific details

of the transformation process whereby the meteorological data is converted into energy quantities.

The difference in performance between the metrics is intuitively consistent with the complexity of the forecast information: ACC essentially depends only on the first moment of the forecast ensemble distribution (i.e., its mean value), whereas the CRPSS is sensitive to higher moments of the forecast ensemble distribution (i.e., its variance, skew and kurtosis). It is worth noting, that these differences in skill may impact some users differently to others (e.g., previous research suggests that an user

seeking profit maximization based on purely speculative trading may achieve it with a skillful forecast of the ensemble mean alone, whereas more advanced risk management strategies may require skill in forecasting higher moments of the forecast distribution's shape; see Lynch (2017) for more details).

This discussion of the metric-dependence of forecast skill shows that the user should use a metric appropriate for their particular application at a particular lead time. For example in week 1 it may be appropriate to use CRPSS to get a forecast of

potential demand, whereas in week 2 or higher using the ACC to get an indication of potential high or low demand-net-wind behaviour is more appropriate. The nature of an appropriate skill metric will, however, very much depend on the decision-making process the forecast seeks to support (e.g., trading, maintenance scheduling, or anticipation of extreme events).

Having highlighted countries where forecast skill is consistently greater or less than others in Figure 1 and 2, we might hope that there is a regional effect whereby clusters of neighbouring countries show enhanced or reduced skill. However,

examining skill maps such as Figure 3 reveals little sign of this. Although there is a hint that Northern and Eastern Europe have enhanced skill at longer lead times compared to Southern Europe when multiple metrics are compared (see Figure 3c,g and k for enhanced skill in week 3).

### 3.4 Case studies

Examining skill scores is very useful for understanding the long-term potential for forecast improvements compared to tradi-

tional methods (such as a past reanalysis climatology). However, it can also be useful to look at some forecasts from memorable events to see how the potential decision making could have been informed. In the following subsections we examine the forecasts for two extreme events to give an indication of the lead time at which sub-seasonal forecasts can provide clear indications of upcoming events. In doing so, we chose to subjectively focus on two forecast criteria: firstly, the extent to which the forecast ensemble suggests an shift towards the tercile in which the observation lies; and, secondly, whether or not any individual

ensemble members captured the intensity of the event observed. It is important to note these case studies are included to provide an illustrative counter-factual discussion of how a potential decision-maker might have benefited (or not) in these specific events. The performance of the forecasts in these events does not, however, constitute a skill assessment, and the previously discussed skill scores should be referred to for information on general predictability.





### 3.4.1 Prolonged cold spell of December 2009

The first case study shown in Figure 4a is from December 2009, a period where the United Kingdom experienced exceptionally cold weather, with temperatures down to -18°C on the 28th and 29th December, with monthly mean temperatures 1.6°C below average (Prior et al. (2011)). These were the lowest recorded temperatures of the winter. This is a time of year where demand is generally suppressed due to human behaviour over the Christmas holidays. However, the demand modelling framework used here only accounts for fluctuations in the temperature dependent component of demand (see section 2.3). Anomalous signals

in this would be predicusing temperature forecasts as an input and could be useful for grid management over the Christmas period.

In Figure 4a a purple star shows the weekly-mean demand for the week of 26th December - 1st January, the forecast week including these anomalously cold days. The event is exceptionally cold compared to the climatology, with a demand anomaly in the 90th percentile. Figure 4a shows forecasts of this event from one to four weeks lead time in the ECMWF model. In

weeks 4, the signal was for a low demand event. However from 3 weeks lead time there is the beginning of the signal for a cold event (i.e. the upper tercile is the most likely outcome), which intensifies as it gets closer (with several ensemble members showing outcomes of comparable or stronger intensity). At 2 weeks lead time 41% of members are predicting an event in the top decile of demand, with 26% of members also showing this at 1 weeks lead time, with a 72% probability of an event in the upper tercile. Overall, it is therefore possible to conclude that this particular forecast provided a strong indication of events in

advance and that this information on the potential for increased demand may have been useful for system planning.

### 3.4.2 Storm Anatol, 2nd-6th December 1999 high wind event

The second case study event shown in Figure 4b is for the demand-net-wind over in Germany for the week of 2nd - 8th December 1999 which was the period during which storm Anatol passed over Europe. Forecasts from the ECMWF model are shown. Although this storm is mostly known for the large amount of loss and damage caused across central-Europe (Roberts

et al. (2014)) the passing of Extra-tropical cyclones also has the potential for large amount of wind power generation, which could suppress demand-net-wind.

Figure 4b shows the observed demand-net-wind is in the 10th percentile. At 4 weeks lead time the predicted distribution of weekly-mean demand-net-wind is skewed towards the upper tercile, suggesting a high demand-net-wind event (demand-net-wind in the upper tercile) is more likely to occur. At 2-3 weeks lead time the distributions are more similar to the climatological

distribution. However, there is much more uncertainty in week 2, with a 24% chance of a high demand-net-wind event forecast. By week 2 there is only a 6% chance of a extreme low demand-net-wind event (an event in the 10th Percentile), which increases to 19% in week 1. In week 1 there are no ensemble members forecasting the upper-tercile of demand-net-wind, so the forecast is quite confident at this lead time of a low demand-net-wind event (with 84% of members forecasting the lower tercile). However, this forecast offered little benefit at lead times beyond lead week 1 (days 5-12) and had the potential to hinder

decision making in lead week 4.



## 4 Conclusions

This study has assessed the skill of a new dataset of forecasts of European national demand, wind and solar power generation. A comprehensive skill assessment is not the goal of this paper but, nevertheless, multiple skill metrics have been used to illustrate some of the many and varied potential applications of the sub-seasonal forecasts with increasing lead time. It is clear

that the forecasts offer a high level of skill in week 1, showing that sub-seasonal forecasts can provide useful information when averaged over days 5-11 of the forecast. At longer lead times skill decreases and, though skill is clearly present in many different metrics, it is not necessarily present at all times, variables and locations. A particularly important point is that depending on the users chosen application, a different skill metric and lead time will be most relevant from this work, and users should therefore pay close attention to assessing the qualities of the forecast in relationship to the decisions they wish to make.

It is hoped that the distribution of the dataset (and the initial assessment of some of the broad characteristics of forecast skill presented here) will encourage the exploration of sub-seasonal forecasts within the energy research community. Through the extensive bias correction of meteorological data, we hope this dataset can be used as an introductory resource for those wishing to investigate if sub-seasonal forecasts could be used for their particular energy application.

Differences in predictability between countries may arise from a mixture of sampling uncertainty, fundamental differences

in the predictability of different geographical regions, or as a result of complex sensitivities introduced by the conversion from weather to energy variables. An in-depth discussion of the causes of the detailed structure of spatial differences in forecast skill is beyond the scope of this study, but their presence emphasises the need for a greater process-based understanding of both the meteorology and "energy conversion" (and consequent decision making) aspects of the forecasting process.

In summary, sub-seasonal predictability is present in many aspects of energy demand, wind power and solar power gen-

eration, which could provide useful information for decision making multiple weeks ahead. We hope that the dataset will provide energy-researchers who are not necessarily specialists in climate prediction, the opportunity to explore the uptake of sub-seasonal forecast data within the energy sector.

## 5 Data availability

The matching reforecast and reanalysis pairs at 1.5° spatial resolution used in this study from 1996-2016 for ECMWF and

1999-2010 for NCEP are currently available at https://researchdata.reading.ac.uk/275/ with assigned DOI: http://dx.doi.org/10.17864/1947.275 (Gonzalez et al., 2020) which will become activated when the paper is accepted for publication. The corresponding national-average 2m temperature, and surface solar radiation variables as well as gridded 100m wind speeds are also available from this source. Hourly, native resolution ERA5 based reanalysis time series of demand, wind power and solar power from 1979-2019 are available at https://researchdata.reading.ac.uk/272/ with assigned DOI: http://dx.doi.org/10.17864/

1947.272, (Bloomfield et al., 2020a) which will become activated when the paper is accepted for publication.





## Appendix A: Demand Model Description

The demand is modelled in three steps. Firstly, for each country, a multiple linear regression is established linking weather properties from ERA5 (Hersbach et al., 2020) to observed national-aggregate daily "total load" (ENTSOE, 2018). The years 2016 and 2017 are used for training. The regression model takes the of Equation A1.

$$Demand(t) = \alpha_0 + \alpha_1(t) + \alpha_2 HDD(t) + \alpha_3 CDD(t) + \sum_{i-4}^{10} \alpha_i DAY(t) \tag{A1}$$

Here t is the time step (in days since 1st Jan 2016) and $\alpha_i$ are the regression coefficients. The first two terms ($\alpha_0$ and $\alpha_1$) correspond to a constant "background" level of demand which permits slow changes over time due to exogenous social, economic and technological factors (e.g., Gross Domestic Product, population, energy efficiency and distribution-level generation). The final six terms ($\alpha_4$ to $\alpha_{10}$) correspond to a series of dummy variables representing the day of the week. That is, the function is
equal to 1 if day corresponds to a day of type (where $i = 1$ is Monday, $i = 2$ is Tuesday, ..., $i = 7$ is Sunday) and zero otherwise. The remaining two terms ($\alpha_2$ and $\alpha_3$) indicate the weather sensitivity of demand corresponding to heating and cooling degree days (HDD or CDD) as shown in Equations A2 and A3

**if** $T(t) < 15.5$ then $HDD(t) = 15.5 - T(t)$ $\qquad$ **else** $HDD(t) = 0$ $\tag{A2}$

**if** $T(t) > 22$ then $CDD(t) = T(t) - 22$ $\qquad$ **else** $CDD(t) = 0$ $\tag{A3}$

T is the country-average daily-mean temperature, calculated over all reanalysis grid boxes using country masks. It is noted that, formally, demand represents a unit of energy volume rather than an energy rate (i.e., corresponds to joules or, equivalently, rather than watts or GWday). Here, however, all analysis is performed on daily time steps so GWday and GW are numerically identical and so, following convention in the power systems literature, demand values are referred to in GW. For consistency, a similar convention is applied to HDD and CDD (i.e., the formal unit of °Cday is expressed as °C).
In Equation A2 and Equation A3 the thresholds match those used by the European Environment Agency (see, e.g., Spinoni et al. (2018)). A country's HDD or CDD time series is zero if T(t) is between 15.5 °C and 22 °C as this is the temperature range in which demand is not believed to be weather-sensitive. Each country has a unique regression model, where any combination of terms can be chosen, such that the Akaike Information Criterion (Wilks, 2011) is minimised). A full set of the regression coefficients (by country) can be found on the Reading Research Data Repository (see data availability section).
Once the regression parameters are established for each country, the second phase is to apply the regression model over the entire historic period in the ERA5 dataset from 1979 to 2019. In this step, it may be desirable for some users to remove the confounding socio-economic behaviours that tend to obscure the connection between weather and demand, thereby "normalising" the demand to a fixed background level and removing day-of-the-week effects. To this end, two version of the model are



ouput, one with the full demand (including all parameters) and one weather-dependent demand, where $\alpha_1$ and $\alpha_4$ to $\alpha_{10}$ are
set to zero, and $\alpha_0$ set to the value of $\alpha_0 + \alpha_1(t)$ at the start of 2017.

Once the forty years of daily-mean demand are created for all of the 28 European countries, the final step is to convert the
data to hourly temporal resolution, as this is desirable for a number of energy-meteorology studies involving power system
simulation. The daily-mean demand data is down-scaled to hourly resolution using a prescribed diurnal cycle. A different
diurnal cycle is determined for each meteorological season based on the recorded 2016-2017 demand data available from
ENTSOE (2018), see Figure A1 for an example of these for Austria. Each daily demand value is down-scaled to hourly
resolution using a linear combination of relevant diurnal curves (e.g., the daily-mean demand for 1st December is down-scaled
using a 50–50 weighting of the diurnal curves derived from the autumn (September to November) and winter (December to
February) hourly data). This method was also used in Bloomfield et al. (2016).

The resulting $R^2$ and root-mean-square error scores for the full hourly model runs are included with the available hourly
modelled data (Bloomfield et al., 2020a). The average country $R^2$ is 0.92 with 8% error. We note that in some countries the
total load data from the ENTSOe transparency platform could include embedded wind and solar power generation (this can
potentially be seen in the summer diurnal cycle curve of Figure A1). This embedded generation could influence the model
verification. However, unpacking this further is beyond the scope of the current modelling framework.

**Appendix B:  Wind Power Model Description**

Wind power is estimated using a physically-based model driven by hourly 100m wind speeds from the ERA5 reanalysis. Prior
to describing the wind-power calculation, it is first noted that the ERA5 100m wind speeds display substantial mean-biases
compared to leading wind-resource assessment datasets such as the Global Wind Atlas (as previously discussed in Bloomfield
et al. (2020b)). A mean bias correction is therefore applied on a grid-point basis prior to conversion into wind power, as small
initial wind speed biases can lead to large differences in wind power generation due to non-linearities in the chosen wind
turbine power curves.

To calculate country-aggregate wind power generation it is necessary to make assumptions about the type of turbines in-
stalled: different turbines respond differently at different wind speeds and the most suiturbine is usually selected as part of the
wind farm commissioning process. Here, three representative power curves are considered, representing type 1, 2, and 3 wind
turbines in the International Electrotechnical Commission wind speed classification (as used in Lledó et al. (2019) and shown
in Figure B1). To select the appropriate power curve, at each grid box a turbine class is assigned based on the 1979-2019 mean,
bias corrected, 100m wind speed. A single estimate of wind power capacity factor (CF) is therefore produced for each grid-box
based on the allocated turbine as shown in equation B1

$$CF(t) = \frac{generation(t)}{maximum possible generation(t)} \tag{B1}$$

The gridded wind power CFs are then weighted by the installed wind power capacity in each grid box as a fraction of the
national total (based on *thewindpower.net* online database). Finally, the gridpoint CF values are averaged over the country

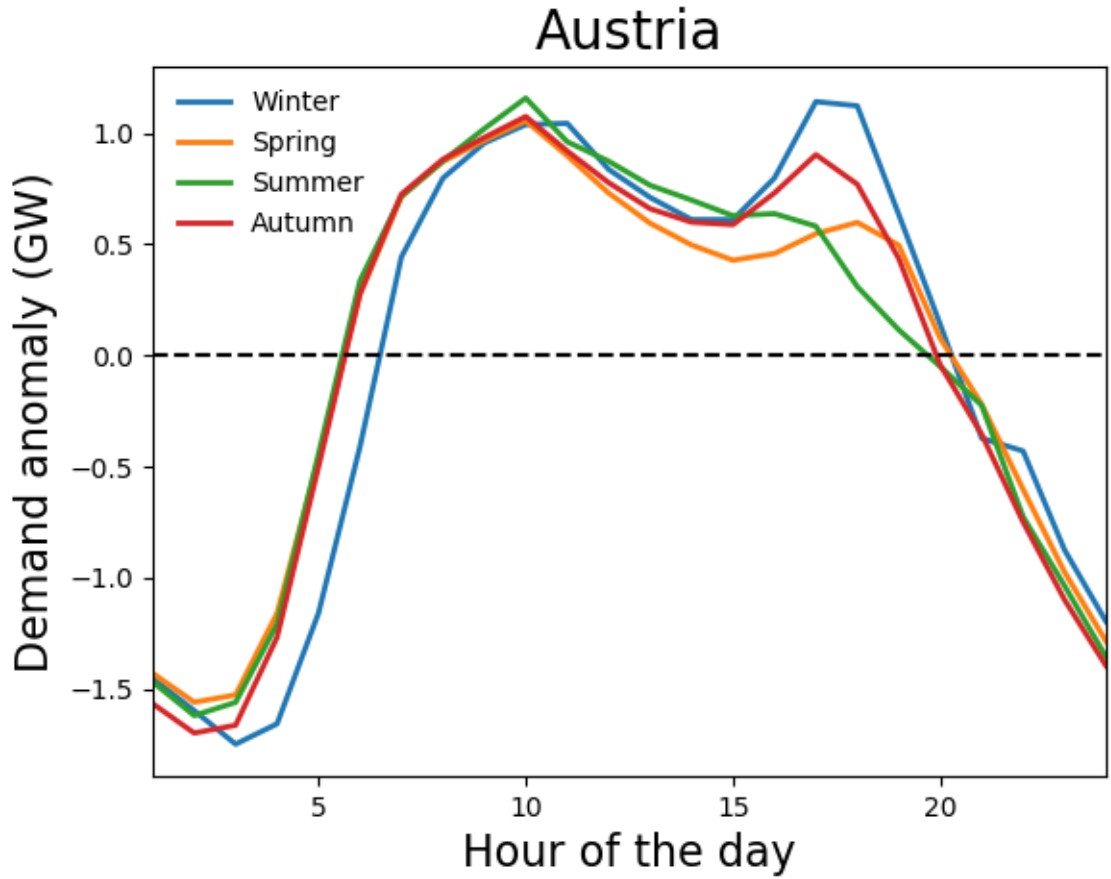

**Figure A1.** Composite diurnal demand cycles for each meteorological season of the year, expressed as an anomaly from the mean daily demand. Seasons are defined as Spring (March–May), Summer (June–August), Autumn (September–November) and winter (December–February.) created from the average of the 2016–2017 demand data from ENTSOE (2018).

domain (weighted by the total installed generation in each grid box) to produce a hourly time series of country-aggregated wind power CF. This has been created for the 28 countries which data is available from the ENTSOe transparency platform (ENTSOE (2018)). Typically, this CF is then multiplied by the 2017 installed capacity to produce daily national-total wind power production. Alternatively, the CF can be multiplied by an installed capacity scenario (such as a trebling of existing 405 capacity) if a larger amount of wind power generation is required. Where good quality data are available from ENTSOe for the verification, the technique performs well, consistent with other studies which follow this now-standard overall approach (Sharp et al. (2015), Cannon et al. (2017), Lledó et al. (2019)) with average $R^2$ of 0.89, and average percentage error of 11%.



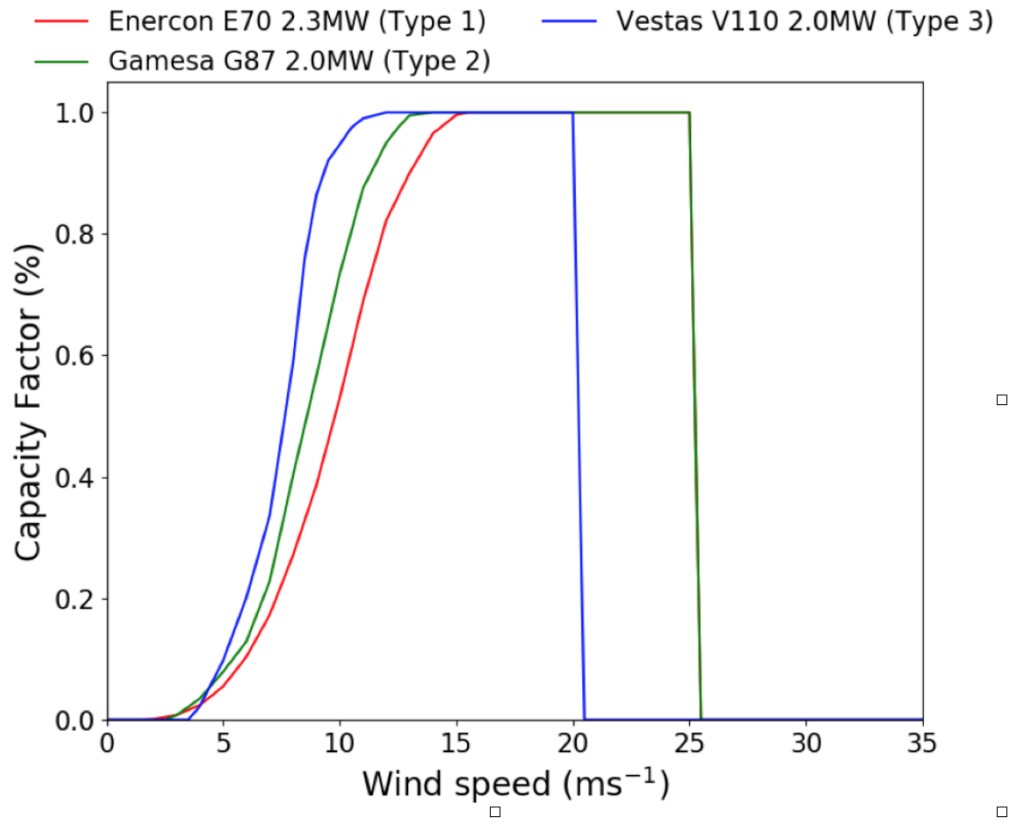

**Figure B1.** Wind power curves used in wind power model.

## Appendix C: Solar Power Model Description

Recent years have seen a rapid increase in installations of solar PV modules. These are associated with a wide range of
different types of PV panel, each with particular weather-response characteristics. This model presents a simple but effective
national-aggregate capacity factor estimate, representing a compromise across many different types of PV panel (for which the
specific details, properties, and even installation locations are unknown). The model is based upon one taken from Evans and
Florschuetz (1977) which depends only on near surface air temperature and incoming surface solar radiation. The solar power
model calculates capacity factor (CF) in each reanalysis grid box at each time step from Equation C1.

$$CF(t) = \frac{power}{power_{STC}} = \eta(G,T)\frac{G(t)}{G_{STC}(t)} \tag{C1}$$





Where G is the incoming surface solar radiation and T is the grid box 2m temperature, and t is the time step (days). STC stands for standard test conditions (T = 25°C G = 1000 Wm$^{-2}$) and $\eta$ is the relative efficiency of the panel following Equation C2.

$$\eta(G,T) = \eta_r[1 - \beta_r(T_c - T_r)] \tag{C2}$$

Where $\eta_r$ is the photovoltaic cell efficiency evaluated at the reference temperature $T_r$, $\beta_r$ is the fractional decrease of cell efficiency per unit temperature increase and $T_c$ is the cell temperature (assumed to be identical to the grid box temperature). This original model is designed for the calculation of solar power yield from a specific panel, rather than over a large grid-box area as is present in a reanalysis, and also represents a rather dated view of modern PV cell performance. As such, their indicated value of yields very low capacity factors using re-analysis and evaluated against power system records. It is therefore

necessary to derive a new estimate of the relative efficiency of a panel at standard test conditions. To do so, a complex empirical relationship for is first considered following Bett and Thornton (2016). $\eta_r$ varies with irradiance and temperature but the change in $\eta_r$ is modest for irradiances in excess of 100 Wm$^{-2}$. This complex dependency can therefore be simplified to an average value (averaged over irradiances 0-1000 Wm$^{-2}$) at the standard test conditions temperature (25 °C, see dashed line in both panels of Figure S4) with only a modest loss in model quality. is thus set to a constant value of 0.90. Similarly, the vertical

distance between the curves on the top panel (corresponding to relative efficiency changes associated with constant intervals of temperature change) is near-constant over a wide range of temperatures and insolations such that is set to constant 0.00042 $°C^{-1}$

Using the resulting model, hourly solar power capacity factor data is calculated at each grid box, using the ERA5 reanalysis 2m temperature and downwelling surface solar radiation. This is converted to a country-aggregate capacity factor assuming a

uniform capacity distribution (over all the grid-boxes in the country). Though crude, this spatial approximation is the only one available as, unlike wind power, there is little good quality information about where solar generation capacity is located. The model performs well with an average daily $R^2$ of 0.93 with 3.2% error for countries where data was available at high enough quality for validation from the ENTSOe transparency portal ENTSOE (2018).

*Author contributions.* Bloomfield developed reanalysis demand, wind and solar power models and wrote the main body of the paper.

Charlton-Perez and Gonzalez applied the methods to the sub-seasonal forecast models. Bloomfield calculated skill scores for the various energy variables. All authors contributed to the manuscript text.

*Competing interests.* No competing interests are present



*Acknowledgements.* This work was conducted as part of the sub-seasonal to seasonal forecasting for energy (S2S4E) project, which was funded by the European Union's Horizon 2020 Research and Innovation Program (grant agreement number 776787). We particularly thank

Llorenç Lledo for the code used to plot the forecast case studies (available in this repository:

https://cran.r-project.org/web/packages/CSTools/vignettes/PlotForecastPDF.html ) Robert Darby for his help making the datasets open access, and to all the S2S4E project partners for useful discussions in the development of these data sets.



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



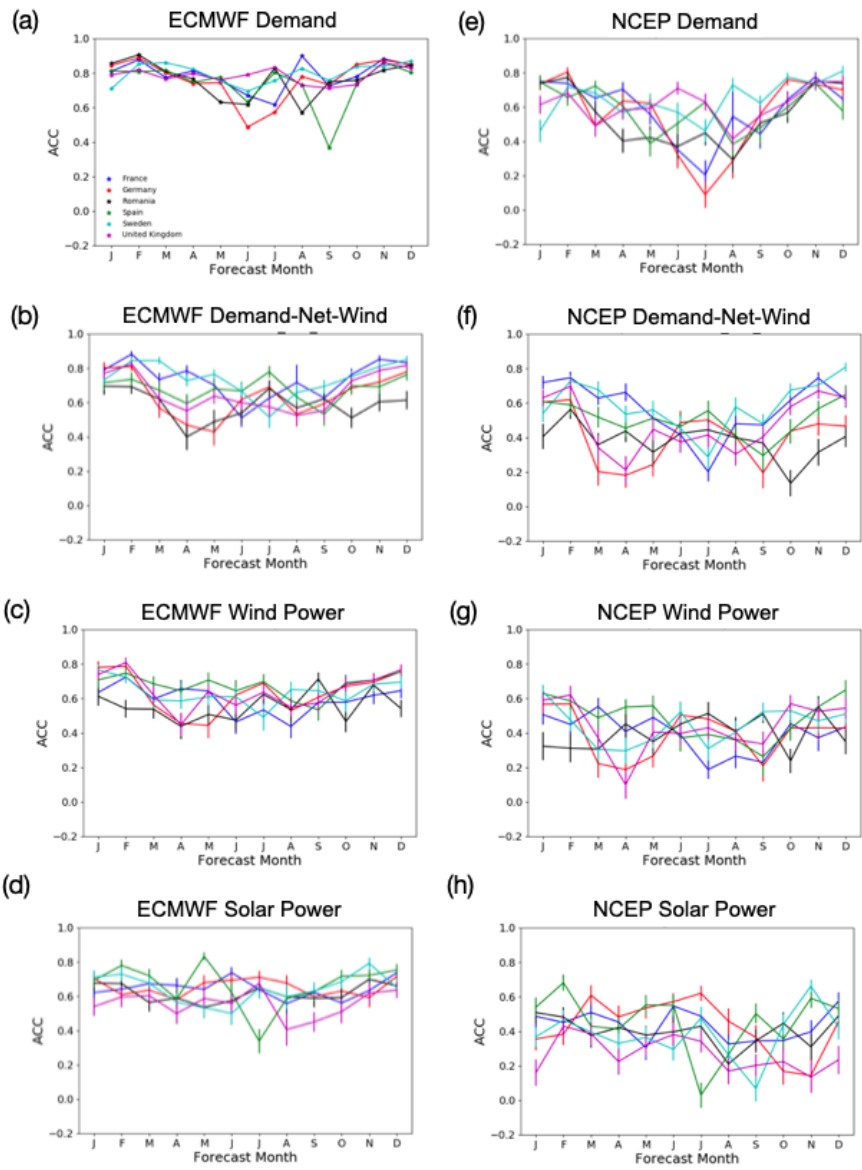

**Figure 1.** Monthly Anomaly Correlation Coefficient (ACC) for sub-seasonal forecasts verified in week 1 (days 5-12) for four energy variables, for the ECMWF (a-d) and NCEP (e-h) sub-seasonal prediction models. Error bars show the 95% significance. Colours show a set of representative case study countries: France, Germany, Romania, Spain, Sweden and the United Kingdom

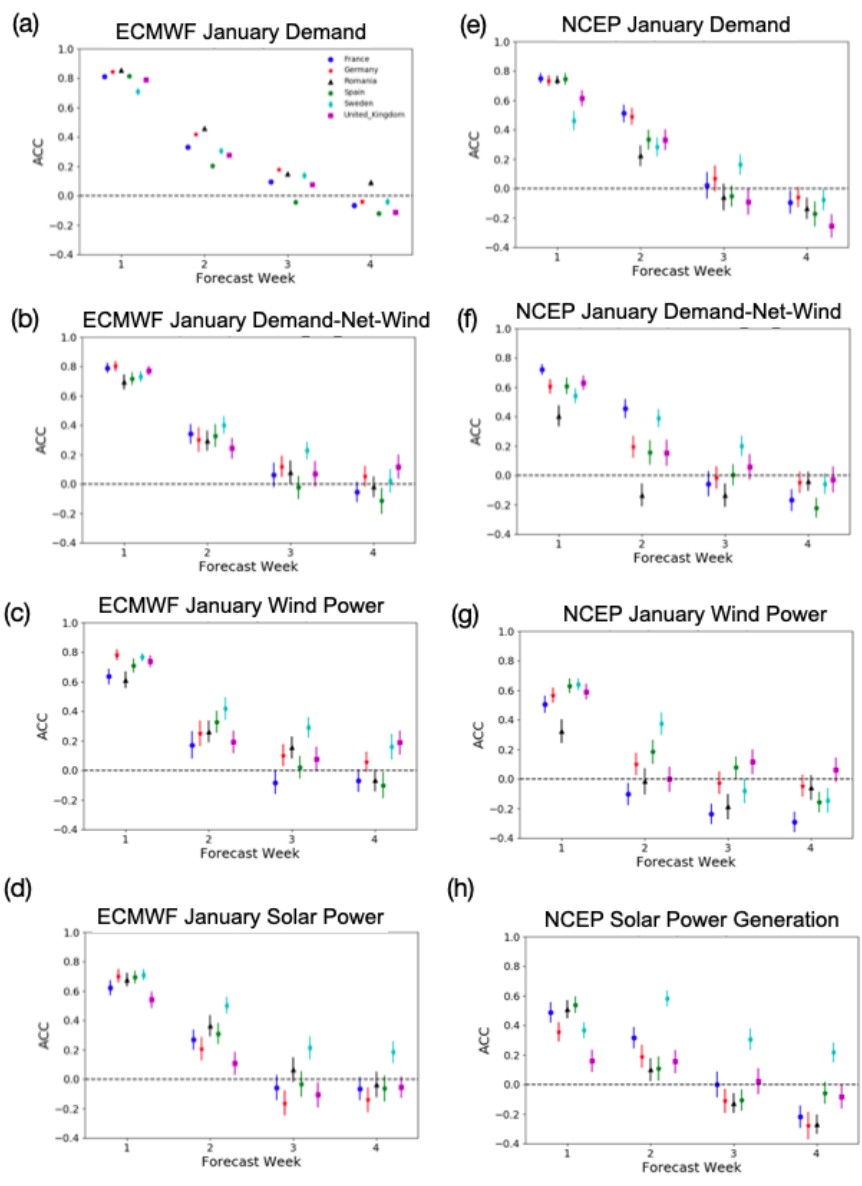

**Figure 2.** Forecasts of weekly-mean January energy-variables for the ECMWF (a-d) and NCEP (e-h) models. Definitions of the lead weeks are given in Section 2. Error bars show the 95% significance. Colours show a set of representative case study countries: France, Germany, Romania, Spain, Sweden and the United Kingdom

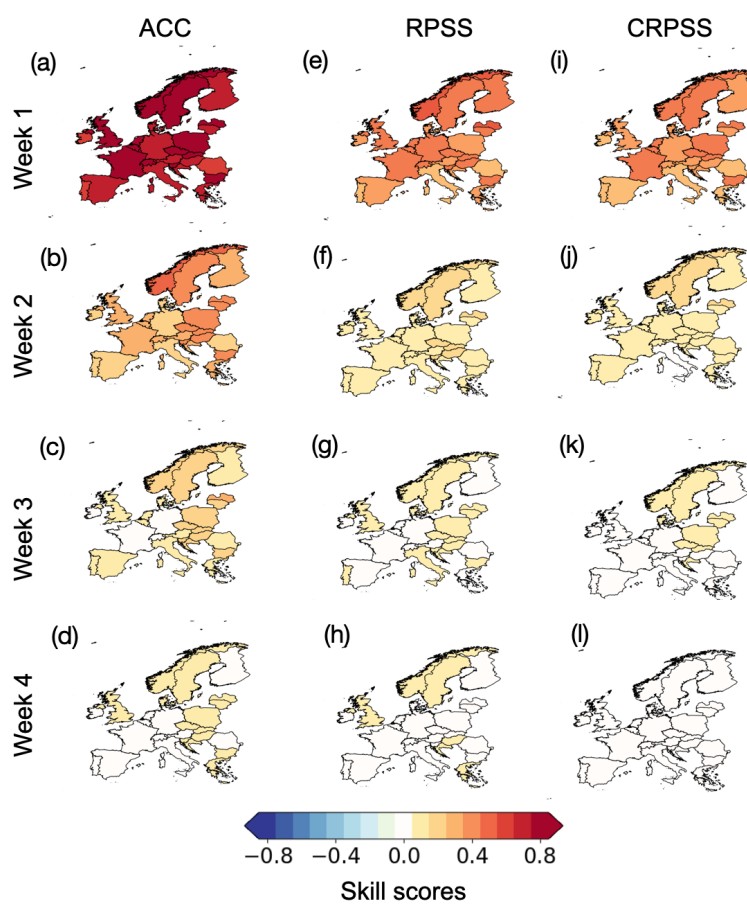

**Figure 3.** Country-level skill scores for weekly mean demand-net-wind from the ECMWF model for (a-d) Anomaly Correlation Coefficient, ACC (e-h) Rank Probability Skill Score, RPSS (i-l) Continuous Rank Probability Skill Score, CRPSS. Definitions of skill scores are given in Section 2.

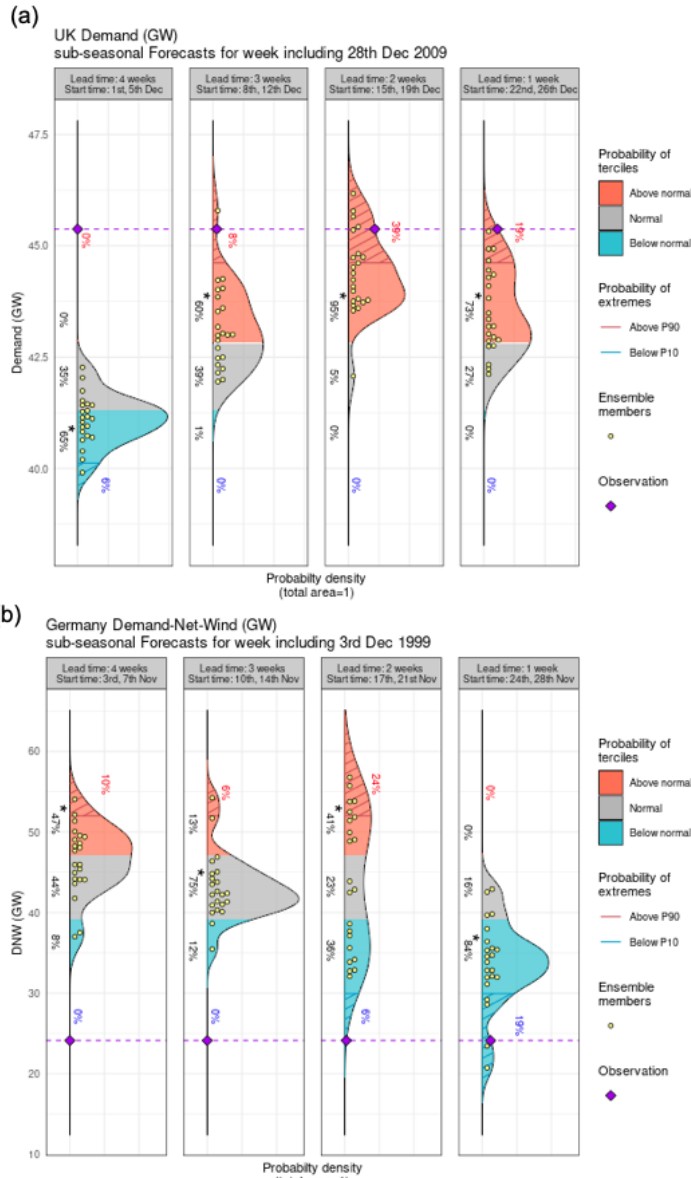

**Figure 4.** Weekly-mean forecasts verifying for the case study events of (a) UK high Demand, December 28th 2009 (b) Storm Anatol 2-8th December 1999. Purple dots give the weekly mean verification from ERA5. From left to right corresponds to forecasts launched from lead weeks 4 to 1. Definitions of the lead weeks are given in Section 2. Shaded areas represent climatological terciles with hatching showing the 10th and 90th percentiles calculated from 1979-2019 in ERA5.