# Peer review of "Sub-seasonal forecasts of demand, wind power and solar power generation for 28 European Countries"

_Earth System Science Data, 2020_

## Author Comment (AC1)

**Reviewer 1:**

The manuscript is written in good english and very well organized. The topic is timely and to my knowledge there exists no comparable study for the time-range aimed at (S2S).

**We thank Reviewer 1 for their positive feedback.**

The reason for this "gap" is that the measurements and weather forecasts are prone to large uncertainties, which is adressed in this manuscript, too. I may add that I am personally very sceptical about a general capability to forecast on a range longer than 1 month with an uncertainty lower than the long-term standard deviation, being from the nonlinear dynamics community . However, with skilled data analysis, there may occur situations which allow a better forecast. So, a systematic approach to clean, and process data is welcome.

We agree that the skill from these sub-seasonal forecast past 1 month is generally low (see Figure 2 showing no remaining skill for some variables past week 3, days 19-25). However we hope this work has shown that not all weather-driven energy variables have the same level skill at extended lead times, so depending on the region and country of consideration skill levels could be variable. Reviewer 2 has also highlighted an important point about windows of opportunity for increased skill at these extended ranges.

In general, the estimate of demand is an issue of national regulations and the authors try to use a general approach to achieve comparability of results. This is very helpful for benchmarking and I appreciate that. The data used are preprocessed and prepared in a useful way, too.

**Thank you**

A few technical comments:

differentiation of the data has not been explained. wrongfully computed differences (as e.g. for the temporal differentiation) can result from too short local intervals. This may not hold, here, though (cf. K Ahnert, M Abel, Computer Physics Communications 177 (10), 764-774).

We have revisited the section where the data is described to confirm that the description of differentiation is clear. We have also checked mathematically that this has been done correctly.

A formulation like "Northern and Eastern Europe have enhanced skill ... compared to Southern Europe" is sematically incorrect, the authors probably mean The forecast skill of .. is higher than. The formulation actually is really funny if taken literally (or may give rise to southern european discomfort).

We have corrected any grammatical errors highlighted by your comments (and throughout the text) to check that everything is correct. Thank you for spotting this particular error.

The skill metrics used are typical, however a full statistical characterisation in terms of moments would maybe be useful. This is still unclear to me, and I have never seen a comparable study.

We agree that no single verification metric (or even set of metrics) can assess all aspects of forecast skill – this motivates the release of the data with the assessment here presented as a "preliminary" assessment of gross skill using common metrics.

The study of extreme events and their predictability is a good idea, but as we understand today, they may well be explained by the instability of the polar vortex. As far as I understand, this cannot be found by the data under consideration, which in turn limits its usefulness for such analyses.

We agree that the study of extreme events is very important, and hope that future work can examine these impacts in a more systematic way than the case studies presented here. The hindcasts contain 20 years (ECMWF) and 12 years (NCEP) respectively, so a study of the predictability and drivers of extreme events could be completed as future work. as well as a discussion of the energy indictors during different polar vortex states, similar to the Bueler et al., study suggested by Reviewer 2.

In general, I highly appreciate the idea to create a consistent dataset for the analysis of ensemble forecasts and their use for power generation. In my opinion, the strength of the paper lies in the data processing, whereas the methodology and metrics used are rather a demonstration of the direction further studies can take with the data set.

The aim here was to demonstrate the general skill levels within the dataset, to promote future collaboration. We agree that the novelty here is the dataset, and hope that it (and the described methods) can be useful to many pursuing scientific studies in the future.

I recommend the manuscript for publication, ideally with some of the remarks addressed.

**Reviewer 2:**

This paper provides a novel data set of daily, national-aggregated sub-seasonal ensemble reforecasts of electricity demand, wind and solar power generation for 28 European countries. This is the first time that S2S reforecasts from ECMWF and NCEP have been thoroughly calibrated and post-processed for the purpose of energy-system modeling and investigations of forecast skill, weather variability, and potential benefits from such forecasts for the energy sector. The paper describes the data and reviews forecast skill for several use cases as a demonstration.

The paper is generally very well-written and suitable for ESS. The provided data will likely foster collaborations between meteorological research and the energy industry in a multitude of applications. I recommend to publish and have only few minor comments.

Thank you for the very positive feedback on the article. We are thrilled you believe it will be useful to develop future collaborations between energy and meteorology and this was our primary aim.

**Minor comments**

- please carefully check for redundancies. I noted few sections which appeared repetitive. I try to indicated these below.

**We have removed any repetition from the article and appendices.**

- I found it a bit confusing that you name the weekly mean of day 5-11 as week 1 and so on. I would call this week 2 as this seems much more common. In the current form just stating e.g. that there is little skill in week 1 and hardly skill in week 2 undermines the potential of subseasonal forecasts as you would rather think of 1-7 and 8-14 day forecasts not 5-11 and 12-18...
- We agree that the definition of the weeks is non-traditional. The definitions are used following on from the sub-seasonal to seasonal prediction for energy (S2S4E) project. This was chosen due to climate services requiring time to process data once it is released from an operational centre. To maximise the usefulness of a 'week 1' forecast that may take a few days to be processed then days 5-11 were chosen. The definition also takes into account that the extended range forecasting systems are not designed to be used for 'weather forecasting' i.e. days 0-5. We therefore exclude this early period. This weekly definition is used in Weigel et al., (2008) and highlighted in Coelho et al., (2019) as a common timescale for verification of S2S forecasts.
- 112: it would help to mention the number of ensemble members upfront and make sure to later stress the differences of reforecast to operational data. 113: It would help to state how many forecasts are provided in total (initial times, and individual realisations (initial times x ensemble members)

We have now included this information in the abstract and made these points clearer in the methods description.

116: "demand-net-renewables" appears a bit jargon for people not familiar with energy meteorology - avoid in the abstract.

Good idea. This has been removed.

-

116: Why do you limit the investigation on winter, stressing different levels of skill in seasons might trigger even more interest.

The limitation to winter is an interesting point. We had limited the discussion in the paper to winter for brevity (and due to this being the season with most promising skill levels). However, increased discussion about the year-around skills levels in Figure 1 are now included, as promising skill levels are seen throughout the year.

117, 152 and later in the paper: It would also have been nice if you had shown some truly windows of forecast opportunities on S2S scales, e.g. contrasting MJO active vs. MJO inactive periods, or periods of enhances / weakened SPV.

We agree that there is the potential to investigate windows of opportunity (e.g. related to modes of atmospheric variability, MJO, ENSO, Sudden Stratospheric warmings.) However, this was beyond the scope of the current study which aimed to give an overview of the available data and a general indication of the skill levels present. This is an excellent topic for future work and we've included this in the discussion, referencing papers by Bueler et al., and Lledo et al., (2020) looking at skill during different stratospheric states and Madden Julian Oscilation phases respectively.

I61: perhaps also define demand-net-solar and demand-net-renewables explicitly.

**Done**

182: "model drift" is jargon here and I doubt that people from outside meteorology / forecasting understand it. Please define what you mean.

**Done**

1157: remove the " at the end of the line.

**Done**

Sections 2.1-2.5. It somehow remained unclear to me if you use daily-averaged data or 00UTC instantaneous. This is because already the description in 170 of daily ERA5 data is not precise. Only in 196 it is stated that you use daily-mean for the S2S reforecast data. Try to state this earlier in the Section. In that regard 1165 "daily" also leaves some room for interpretation. You likely mean "daily sum" here.

We haved clarified any uncertainties in the methods relating to the energy models, forecast structure and timescales of the variables used in the methods. The issue here is that the fields are available on different timescales from the S2S database. daily-mean 2m temperature, midnight 10m wind speeds and daily accumulation of surface solar radiation. Hopefully the amendments should make this clearer.

1218-221: these results should be compared and discussed along Büeler et al. 2020 how showed similar maps but for meteorological variables aggregated over 30 days. Overall I think the potential of your dataset for energy applications should also be discussed in the light of windows of forecast opportunities on S2S time-scales. Again Büeler et al. 2020 would be an example showing the enhanced / reduced skill for several countries following extreme states of the stratosphere though not investigating energy variables, yet. Soret et al. 2019 is another example using S2S forecast in the energy context. Both papers are the only ones so far that I found screening the last 2 years S2S database related publications at http://s2sprediction.net/static/documents#publications .

We have also included reference to the suggested papers and included a section highlighting how these datasets could be used to investigate windows of opportunity in forecasts.

1230: You should briefly explain how to interpret the ACC. Usually only ACC >0.6 are considered as useful forecasts, cf textbook of Wilks.

This is now mentioned in the Verification Metrics section to motivate the later use of RPSS and CRPS.

1237: A bit unclear what "This" refers to. 1238 again I find it odd to talk of week 2 if you mean 19 days.

**This has been clarified.**

1248: Spatial differences might also be due to meteorological weather variability which can be different in different countries.

Good point, this has been clarified and compared to the results in Bueler et al., 2020.

1252: repetition

Removed.

1258-262: Very nice discussion!

Thank you!

1290: why weeks in plural?

**This has been removed**

Section 4: avoid repetitive statements in the Conclusions. The conclusions could stress even a bit more potential applications and also refer to first studies going in that direction, however using meteorological variables.

We have included a paragraph to enhance this discussion and removed repetition.

Section 5 & Appendix A & 1 385, 433: It remains a bit elusive how you used the hourly ERA5 data in this study.

This is a good point. The hourly ERA5 data is available here as a point of comparison (as the data within the forecasts is only for a daily timescale (either daily-mean, daily accumulation or might night, depending on the weather variable). This would allow a user the option to unpack what is 'lost' through having the different resolution weather/energy fields.

1346: only two years of training data seems little. Some justification would be good.

A statement has been included to justify this.

Figure A1: It is a bit surprising why you now show Austria as an example, as other countries were discussed before. I would recommend using one of the exemplified countries here.

Good point, this has been changed for the United Kingdom as this was one of the case study countries.

1416: repetition

Removed

References

Büeler, D., Beerli, R., Wernli, H., & Grams, C. M. (2020). Stratospheric influence on ECMWF subâseasonal forecast skill for energyâindustryârelevant surface weather in European countries. Quarterly Journal of the Royal Meteorological Society.

Soret, A., Torralba, V., Cortesi, N., Christel, I., Palma, L., Manrique-Suñén, A., ... & Doblas-Reyes, F. J. (2019, May). Sub-seasonal to seasonal climate predictions for wind energy forecasting. In Journal of Physics: Conference Series (Vol. 1222, No. 1, p. 012009). IOP Publishing.

References: Weigel, A. P., Baggenstos, D., Liniger, M. A., Vitart, F., & Appenzeller, C. (2008). Probabilistic verification of monthly temperature forecasts. Monthly Weather Review, 136(12), 5162-5182.

Coelho, C. A., Brown, B., Wilson, L., Mittermaier, M., & Casati, B. (2019). Forecast Verification for S2S Timescales. In Sub-Seasonal to Seasonal Prediction (pp. 337-361). Elsevier.